# Enhanced Structural Stability and Electrochemical Performance of LiNi_0.6_Co_0.2_Mn_0.2_O_2_ Cathode Materials by Ga Doping

**DOI:** 10.3390/ma14081816

**Published:** 2021-04-07

**Authors:** Zhibei Liu, Jiangang Li, Meijie Zhu, Li Wang, Yuqiong Kang, Zhaohan Dang, Jiasen Yan, Xiangming He

**Affiliations:** 1Beijing Key Laboratory of Fuels Cleaning and Advanced Catalytic Emission Reduction Technology, School of Chemical Engineering, Beijing Institute of Petrochemical Technology, Beijing 102617, China; 2018520009@bipt.edu.cn (Z.L.); 2018520012@bipt.edu.cn (M.Z.); 2019520039@bipt.edu.cn (Z.D.); 2019520009@bipt.edu.cn (J.Y.); 2Institute of Nuclear and New Energy Technology, Tsinghua University, Beijing 100084, China; 3Graduate School at Shenzhen, Tsinghua University, Shenzhen 518055, China; kang.yuqiong@sz.tsinghua.edu.cn

**Keywords:** lithium ion batteries, cathode material, LiNi_0.6_Co_0.2_Mn_0.2_O_2_, Ga doping, structural stability

## Abstract

Structural instability during cycling is an important factor affecting the electrochemical performance of nickel-rich ternary cathode materials for Li-ion batteries. In this work, enhanced structural stability and electrochemical performance of LiNi_0.6_Co_0.2_Mn_0.2_O_2_ cathode materials are achieved by Ga doping. Compared with the pristine electrode, Li[Ni_0.6_Co_0.2_Mn_0.2_]_0.98_Ga_0.02_O_2_ electrode exhibits remarkably improved electrochemical performance and thermal safety. At 0.5C rate, the discharge capacity increases from 169.3 mAh g^−1^ to 177 mAh g^−1^, and the capacity retention also rises from 82.8% to 89.8% after 50 cycles. In the charged state of 4.3 V, its exothermic temperature increases from 245.13 °C to more than 271.24 °C, and the total exothermic heat decreases from 561.7 to 225.6 J·g^−1^. Both AC impedance spectroscopy and in situ XRD analysis confirmed that Ga doping can improve the stability of the electrode/electrolyte interface structure and bulk structure during cycling, which helps to improve the electrochemical performance of LiNi_0.6_Co_0.2_Mn_0.2_O_2_ cathode material.

## 1. Introduction

With the rapid development of portable electronic products and electric vehicles, higher requirements have been put forward on the energy density, safety, cycle life, and cost of lithium-ion batteries (LIBs). The nickel-rich ternary layered material LiNi_1−x−y_Co_x_Mn_y_O_2_ such as LiNi_0.8_Co_0.1_Mn_0.1_O_2_ (NCM811) and LiNi_0.6_Co_0.2_Mn_0.2_O_2_ (NCM622) exhibit high capacity and low cost, showing a promising application prospect [1,2]. However, with the increase in nickel content, the cycle performance, thermal stability, and safety gradually decrease [3,4]; this happens due to factors such as surface residual alkali, transition metal dissolution, cation mixing, surface irreversible formation of NiO phases, intergranular cracks, and micro-strains [1,2,3,4,5,6,7,8,9]. Among the nickel-rich LiNi_1−x−y_Co_x_Mn_y_O_2_ materials, NCM622 material can be prepared in the air, has a higher lithium ion diffusion coefficient, and has better structural stability [10,11,12]; therefore, it has become the preferred choice for research and commercial application.

In order to improve the electrochemical performance of NCM622 material, some modification methods have been investigated in recent years. One useful method is to coat the surface of NCM622 material with Al_2_O_3_ [13], Co_3_O_4_ [14], SiO_2_ [15], Li_3_PO_4_ [16], Mn_3_(PO_4_)_2_ [17], Li_1.3_Al_0.3_Ti_1.7_(PO_4_)_3_ [18], LiAlO_2_ [19], and Li_2_Si_2_O_5_ [20] in order to improve the stability of the electrode/electrolyte interface and, thus, enhance the capacity, Coulomb efficiency, cyclability, and thermal safety performance. Another effective method is to prepare heterogeneous structural materials. For example, the formation of a heterogeneous interface layer of rock salt phase on the primary particles has been reported to significantly improve the cycle stability of NCM622 material at high temperatures [21]. In addition, element doping is also a very important modification method. Schipper et al. [22] concluded that Zr doping can suppress the phase change of NCM622 from the layered structure to the spinel structure. Liu et al. [23] reported that Mo doping for NCM622 can suppress the loss of lattice oxygen, enhance the cation order, broaden the Li^+^ migration channel, and improve the electrochemical performance under high voltage. Huang et al. [24,25] confirmed that Na and Mg doping can promote Li^+^ migration and improve the rate performance of NCM622. Mofid et al. [26] replaced Co in NCM622 by both Fe and Al, which reduced the degree of cation mixing and improved the cycle stability remarkably. Because F has strong electronegativity, a more stable crystal structure can be obtained while F ions are doped into the oxygen site of NCM622 material. All of the F-doped samples [27], including those co-doped with Na and Mg electrode materials [28,29], showed a good rate performance and excellent cycle performance. Recently, the doping and coating of dual functional modified materials prepared by using the same source, such as PO_4_^3–^ gradient doping and Li_3_PO_4_ coating dual functional materials [30], Zr doping and amorphous Li_2_ZrO_3_ coating dual functional materials [31], have also been reported to greatly improve the cycle performance of NCM622 under high temperature and high voltage.

Ga doping should also be an effective method for stabilizing the layered cathode materials, because the ion radius of Ga^3+^ is close to that of Co^3+^ and Ni^3+^ but has a strong polarization, which is beneficial for increasing the covalence of the O-TM layer in layered lithium transition metal oxides [32]. It has also been proved that Ga doping can significantly improve the structural stability and cycling performance of LiNiO_2_ [33,34], LiCoO_2_ [35], LiNi_0.8_Co_0.2_O_2_ [36], and Li[Li_0.2_Mn_0.54_Co_0.13_Ni_0.13_]O_2_ [37]. However, Ga doping has not been used to improve the structural stability and cycling performance of nickel-rich LiNi_1−x−y_Co_x_Mn_y_O_2_ materials. Although Ga is a dispersed rare metal, its reserves on the earth have reached 1 million tons, which is about 1/7 of those of cobalt, and its price is less than four times that of cobalt [38]; however, since the amount of doped Ga is very small, the increase in cost should be minimal. Therefore, the research on Ga doping to improve the electrochemical performance of nickel-rich LiNi_1−x−y_Co_x_Mn_y_O_2_ materials seems necessary.

In this project, the high-temperature solid-state reaction method was employed to synthesize Ga-doped NCM622 materials. The effect of annealing temperature and Ga content on the structural and electrochemical properties of Ga-doped NCM622 materials was thoroughly investigated. The Ga-doped materials prepared under the optimized synthesis conditions exhibited remarkably improved structural stability and electrochemical performance.

## 2. Experimental

Li[Ni_0.6_Co_0.2_Mn_0.2_]_1−x_Ga_x_O_2_ (x = 0, 0.01, 0.02, 0.03 and 0.05) materials were prepared by the high-temperature solid-state reaction method. Stoichiometric Li_2_CO_3_ (AR, Tianjin Fuchen Chemical Reagent Co., Ltd., Tianjin, China), Ni_0.6_Co_0.2_Mn_0.2_(OH)_2_ (Henan Kelong NewEnergy Co., Ltd., Xinxiang, China), and Ga_2_O_3_ (AR, Shanghai Macleans Biochemical Technology Co., Ltd., Shanghai, China) with cation mole ratio of Li:(Ni + Co + Mn):Ga = 1.08:(1 − x):x were mixed, thoroughly grinded, and then transferred into the furnace (TM-0914P, Beijing Ying’an Meicheng Scientific Instrument Co., Ltd., Beijing, China), where they were preheated at 500 °C for 6 h; they were subsequently calcined for 12 h at 800 °C, 850 °C, and 900 °C, respectively, with a heating rate of 5 °C min^−1^, followed by a cooling down to 500 °C with a rate of 2 °C min^−1^, and then naturally cooling to room temperature.

The structures and microscopic morphologies of Li[Ni_0.6_Co_0.2_Mn_0.2_]_1−x_Ga_x_O_2_ materials were analyzed using an X-ray diffractometer (Rigaku, D/Max-2600-PC, Tokyo, Japan) and field emission scanning electron microscopy (FEI, Quanta-400F, Hillsboro, OR, USA), respectively. X-ray photoelectron spectroscopy (XPS) and energy dispersive X-ray (EDX) spectroscopy analysis for the sample with x = 0.02 were carried out in situ using an X-ray photoelectron spectrometer (PHI, 5000 Versaprobe II, Kanagawa, Japan) and a field emission transmission electron microscope (FEI, Tecnai G2 F30, Hillsboro, OR, USA), respectively. The electrochemical properties of the as-prepared materials were tested by galvanostatic charge–discharge test using CR2032 coin cells, in which the cathode electrodes, comprised of 80% active material, 10% Super P, and 10% poly(vinylidene fluoride) (PVdF), were pasted on porous Al foil; a Li metal chip was used as anode; 1 mol L^−1^ LiPF_6_/EC + DEC + DMC (volume ratio 1:1:1) was used as the electrolyte; and Celgard 2400 membrane (Charlotte, NC, USA) was used as the separator. The coin cells were assembled in a glove box (Etelux Lab2000, Etelux Inert Gas System (Beijing) Co., Ltd., Beijing, China) filled with argon and then installed on a land battery system (LANHE CT2001A, Wuhan Jinnuo Electronics Co., Ltd., Wuhan, China) to test the charge–discharge performance in the voltage range of 2.8–4.3 V. Electrochemical workstation (IM6eX, Zahner Elektrik GmbH & Co. KG, Kronach, Germany) was used to test the electrochemical impedance spectroscopy of the fully discharged cathode electrodes (2.8 V vs. Li/Li^+^), in which the frequency range was 100 mHz–100 KHz, and the AC amplitude was 5 mV. The thermal stability of Li[Ni_0.6_Co_0.2_Mn_0.2_]_1−x_Ga_x_O_2_ materials (typical weight 6.2 mg) in the charged state (4.3 V, vs. Li/Li^+^) was analyzed with a differential scanning calorimeter (Q2000, TA Instruments, New Castle, DE, USA) under the following test conditions: nitrogen atmosphere; heating rate, 5 °C·min^−1^; and temperature range, 50–350 °C.

## 3. Results and Discussion

At first, the calcination temperature for the synthesis of Li[Ni_0.6_Co_0.2_Mn_0.2_]_1−x_Ga_x_O_2_ (x = 0, 0.01, 0.02, 0.03 and 0.05) materials was optimized. Figure 1 shows the XRD patterns of the Li[Ni_0.6_Co_0.2_Mn_0.2_]_0.98_Ga_0.02_O_2_ sample prepared at different calcination temperatures. It is clear that all peaks can be indexed to the layer α-NaFeO_2_ structure with space group R3-m, and no impurity phase appeared. Low cation disorder for all samples can be confirmed by the lattice parameter ratio of *c*/*a* > 4.9 and the ratio of I_(003)_/I_(104)_ > 1.2. With the increase in calcination temperature, the splitting degree of double peaks (006)/(102) and (108)/(110) rose, and the ratio of *c*/*a* gradually increased from 4.9485 for the sample prepared at 800 °C to 4.9548 for the sample prepared at 900 °C, indicating an improved cation ordering. This was further confirmed by the following charge–discharge tests, as shown in Figure 2. Among the three samples, the sample calcinated at 900 °C shows the smallest polarization and the highest discharge capacity. Its discharge capacity at a rate of 0.5 C reached 177.0 mAh g^−1^. In the following work, samples calcined at 900 °C are used to discuss the influence of Ga doping on the structure and electrochemical performance of Li[Ni_0.6_Co_0.2_Mn_0.2_]_1−x_Ga_x_O_2_ (x = 0, 0.01, 0.02, 0.03 and 0.05) materials.

Figure 3 shows the XRD patterns of the prepared Li[Ni_0.6_Co_0.2_Mn_0.2_]_1−x_Ga_x_O_2_ (x = 0, 0.01, 0.02, 0.03 and 0.05) materials. All of the samples still exhibited a well-defined layered structure based on a hexagonal α-NaFeO_2_ structure with low cation mixing between Li^+^ and Ni^2+^ in the lithium layer; no impurity phases emerged even for the sample with x = 0.05. As the Ga content increased, the lattice parameters of *a* and *c* increased gradually from 2.8695 and 14.2202 Å for x = 0 to 2.8763 and 14.2601 Å for x = 0.05, respectively; this occurred due to the fact that the Ga^3+^ ionic radius (0.62 Å) is similar to the average ionic radius of Ni^2+^ (0.69 Å) and Mn^4+^ (0.53 Å) but is larger than that of Co^3+^ (0.545 Å) and Ni^3+^ (0.56 Å). This implies that Ga^3+^ was doped into the crystal lattice successfully. Although the lattice parameters of *a* and *c* improved gradually as the Ga content increased, little change occurred for the lattice parameter ratio of *c*/*a*, indicating that small quantities of Ga^3+^ doping do not generate a negative effect on the two-dimensional layered structure of LiNi_0.6_Co_0.2_Mn_0.2_O_2_ materials.

FESEM images of the prepared Li[Ni_0.6_Co_0.2_Mn_0.2_]_1−x_Ga_x_O_2_ (x = 0, 0.01, 0.02, 0.03 and 0.05) materials are shown in Figure 4. It can be seen that all five samples present the morphology of dense agglomerated secondary spherical particles with a particle size of ~10 µm. It is noted that the primary particle size increased from ~300 nm for x = 0 µm to ~1 µm for x = 0.02–0.05; the crystal planes and grain boundaries became clearer, indicating that a moderate Ga doping is beneficial for promoting crystal growth.

Furthermore, X-ray photoelectron spectroscopy (XPS) and energy dispersive X-ray (EDX) spectroscopy analysis were carried out for the sample with x = 0.02. As shown in Figure 5a, the photoelectron peak of Ga 3d can be observed in the XPS survey spectra. It can be seen from Figure 5b that the binding energy corresponding to the Ga 3d_5/2_ peak is 20.8 eV, which is consistent well with the reported data for Ga_2_O_3_ [39], indicating that the valence state of the doped Ga ions remains +3. Figure 6 presents the EDX mapping of the Li[Ni_0.6_Co_0.2_Mn_0.2_]_0.98_Ga_0.02_O_2_ material. As in the case of Ni, Co, Mn, and O atoms, the doped Ga element is also uniformly distributed, which can conform uniform Ga doping in the material.

To investigate the effect of Ga doping on the electrochemical performance of Li[Ni_0.6_Co_0.2_Mn_0.2_]_1−x_Ga_x_O_2_ materials, the as-prepared materials were assembled into 2032 coin cells for the charging and discharging tests. Figure 7 shows the initial charge–discharge curves (Figure 7a) and the cycling performances (Figure 7b) at 0.5 C (1 C = 200 mA g^−1^) in the voltage between 2.8 and 4.3 V. The discharge capacity of the samples with Ga content x = 0, 0.01, 0.02, 0.03, and 0.05 were found to be 169.3, 170.1, 177.0, 171.3, and 163.2 mAh g^−^^1^ in the first cycle, and 140.2, 150.4, 158.9, 147.3 and 146.9 mAh g^−1^ after 50 cycles, with the capacity retention of 82.8, 88.4, 89.8, 86.0, and 89.7%, respectively. Enhanced discharge capacity for the sample with x = 0.02 may be attributed to the improved cation order. However, when the Ga content was x ≥0.03, the discharge capacity dropped again, which may be due to the excessive doping of Ga^3+^ without electrochemical activity. In addition, the rate performance of Li[Ni_0.6_Co_0.2_Mn_0.2_]_1−x_Ga_x_O_2_ was also investigated, as shown in Figure 8. Among the five samples, the one with x = 0.02 still showed the best electrochemical performance, with a discharge capacity that reached 183.4 mAh g^−1^ at 0.2 C and 121.1 mAh g^−1^ at 5 C, respectively. It was noted that the capacity from 26 cycles to 30 cycles at 0.2 C, followed by a charge–discharge at 5 C from 21 to 25 cycles, still retained 97.9% of its initial discharge capacity at 0.2 C in the first cycle. The above results reveal that a Ga substitution for 2% of the transition metal elements in the NCM622 material can significantly improve the electrochemical performance.

Electrochemical impedance spectroscopy (EIS) analysis was used to further clarify the mechanism for the improvement of the electrochemical performance of NCM622 by Ga doping. The Nyquist plots of Li[Ni_0.6_Co_0.2_Mn_0.2_]_1−x_Ga_x_O_2_ electrodes are displayed in Figure 9. The Nyquist plots consist of the following: (a) a semicircle in the high frequency range assigned to surface film resistance; (b) another semicircle in the medium frequency range assigned to charge transfer impedance; and (c) a sloped line in the low-frequency range assigned to the impedance of diffusion of lithium ions [40]. The analysis of the plots was performed by fitting the equivalent circuit; the fitting results for Li[Ni_0.6_Co_0.2_Mn_0.2_]_1−x_Ga_x_O_2_ electrodes after three cycles are shown in Table 1. Compared with pristine electrodes, the Ga-doped cathode electrodes with x = 0.01, 0.02 and 0.03 exhibited minor decreased surface film resistance (R_f_), and remarkably reduced charge transfer resistance (R_ct_). This indicates that Ga doping can improve the electrochemical activity in the interface of electrode/electrolyte, thereby helping to reduce electrochemical polarization and enhance the capacity and rate performance. However, the R_f_ value for the sample with Ga content x = 0.05 increased significantly, which may be due to the presence of impurity phases on the surface of the material. Although impurity phases are not observed in Figure 1, the research based on synchrotron XRD analysis confirmed the limited solubility of Ga in the LiNiO_2_ and the formation of impurity phase Li_5_GaO_4_ for the Ga-doped LiNiO_2_ samples [34]. The Li^+^ diffusion coefficient can also be calculated by using the method in reference [28] to process the EIS data. The obtained Li^+^ diffusion coefficients for the samples with Ga content x = 0, 0.01, 0.02, 0.03, and 0.05 are 1.30 × 10^−12^, 5.80 × 10^−11^, 8.85 × 10^−11^, 6.92 × 10^−11^, and 3.81 × 10^−11^ cm^2^ s^−1^, respectively. The change trend of the Li^+^ diffusion coefficient is completely consistent with that of the charge transfer impedance, which may be related to the synergy of charge transfer. Compared with the pristine sample, the Ga-doped sample with x = 0.02 exhibited a Li^+^ diffusion coefficient as high as about 53 times, which is consistent with its best electrochemical performance. EIS analysis of Li[Ni_0.6_Co_0.2_Mn_0.2_]_0.98_Ga_0.02_O_2_ was further conducted after 50 cycles, and the results, compared with those of a pristine electrode, are presented in Figure 9b. The R_f_ and R_ct_ of the pristine electrode increased from 18.62 and 32.34 Ω in the 3rd cycle to 96.26 and 135.6 Ω in the 50th cycle, respectively, resulting in an increase in the interface impedance by 180.9 Ω. However, the R_f_ of the Ga-doped sample with x = 0.02 increased by 19.31 Ω, and the R_ct_ decreased by 17.82 Ω due to the improved electrode activation, resulting in an increase in the total interface impedance by only 1.5 Ω after 50 cycles. It can be concluded that Ga doping can effectively suppress the increase in interface impedance of the Ga-doped cathode electrode during cycling, which is beneficial for enhancing capacity retention.

For LiNi_1−x−y_Co_x_Mn_y_O_2_ materials, the increase in interface impedance during cycling is mainly ascribed to the electrode/electrolyte interface side reaction (e.g., electrolyte decomposition, formation of SEI (solid electrolyte interface) film, and transition metal dissolution) and structural instability (e.g., cation mixing, NiO phase formation, microcracks, and microstrains caused by phase transition). The interface side reactions are often coupled with structural instability. For example, new interface side reactions occur on the generated microcracks, which can accelerate capacity fading during cycling [1,2]. Therefore, improving structural stability is very important in order to enhance the cycle life of LiNi_1−x−y_Co_x_Mn_y_O_2_ materials. In this work, in situ XRD analysis was performed on the Ga-doped sample with x = 0.02 during the charge–discharge process, the results of which are shown in Figure 10. As shown in Figure 10a,b, the hexagonal phase structure is maintained all over the charging process—i.e., without obvious diffraction peaks ascribed to the monoclinic phase or the two-hexagonal phase—which is similar to that of LiNi_0.98_Ga_0.02_O_2_ material [33]. This should be attributed to the stabilizing effect of Ga doping on the layered structure.

It can be seen from the refined lattice parameters (Figure 10c) that the *c* value, which represents the interslab distance of the Li layers, increased initially with the deintercalation of Li^+^ ions due to the increase in Coulomb repulsion. However, when most of the Li^+^ ions were removed (corresponding to Li content x = 0.47 in Li_x_MO_2_), a contraction of interslab distance occurred with a decrease in *c* value, which also led to a sharp contraction of *c*/*a* and *V*. For Ni-rich ternary materials, the severe change in the crystal lattice along the c-direction during the charging–discharging process presumably causes the instability of the lattice structure [5,34]. As shown in Figure 10d, the relative changes in lattice parameters of Li[Ni_0.6_Co_0.2_Mn_0.2_]_0.98_Ga_0.02_O_2_ are compared with the reported values of undoped NCM622 material [5] during the delithiation process. It can be seen that the rapid contraction occurred for NCM622 material, while the Li content y in Li_x_MO_2_ was less than 0.45, and the lattice parameters *c*, *c*/*a*, and unit cell volume *V* shrunk by 1, 0.82, and 1.33%, respectively, until the Li content y value reached 0.3 (4.32 V vs. Li/Li^+^) [5]. However, for the Ga-doped sample Li[Ni_0.6_Co_0.2_Mn_0.2_]_0.98_Ga_0.02_O_2_, the shrinkage of lattice parameters *c*, *c*/*a*, and *V* in almost the same delithiation range decreased to 0.57, 0.6, and 0.72%, respectively. In addition, the total unit cell volume of Li[Ni_0.6_Co_0.2_Mn_0.2_]_0.98_Ga_0.02_O_2_ decreased by 1.94% in the entire charging process (4.3 V vs. Li/Li^+^), which is smaller than the 2.6% of NCM622 material [5]. The above results confirm that Ga doping helps to increase the structural stability, which in turn can decrease the intergranular cracks and mechanical strain occurring in cycling, thus improving the cycle life of NCM622 material.

Nickel-rich ternary cathode materials in a deeply delithiated state always suffer from lattice oxygen loss and phase transition at 150–300 °C, accompanied by higher heat generation. Therefore, the thermal stability of nickel-rich cathode materials is a critical aspect for ensuring the safety of lithium rechargeable batteries. Considering this aspect, the thermal stability of Li[Ni_0.6_Co_0.2_Mn_0.2_]_1−x_Ga_x_O_2_ (x = 0, 0.02) materials was tested after being charged to 4.3 V, the differential scanning calorimeter (DSC) profiles of which are shown in Figure 11. The exothermic peak of LiNi_0.6_Co_0.2_Mn_0.2_O_2_ appeared at 245.13 °C, and the total generated heat was 561.7 J·g^−1^. However, for the Ga-doped sample Li[Ni_0.6_Co_0.2_Mn_0.2_]_0.98_Ga_0.02_O_2_, in addition to an exothermic peak attributed to the phase transition at 271.24 °C, a small exothermic peak also appeared at 292.41 °C, which may be related to the side reaction between the cathode and electrolyte [41]. It is clear that not only the exothermic temperature increases, but the total exothermic heat also decreases to 225.6 J·g^−1^. This indicates that Ga doping is beneficial for improving the thermal stability of NCM622 material. We believe that this is ascribed to the enhanced structural stabilization by Ga doping.

Table 2 lists the performance of some doped LiNi_0.6_Co_0.2_Mn_0.2_O_2_ materials reported in recent years. These materials were selected because they have the same charge cut-off voltage, which is convenient for performance comparison. It can be seen that the Ga-doped sample synthesized in this work exhibits the highest discharge capacity and rate performance. While comparing the cycle performance, the capacity retention of some samples seemed to be somewhat higher. However, since these were obtained under the circumstances of high rate (1 C) and low initial capacity, the comparison was not feasible. In addition, the Ga-doped sample showed significantly improved thermal stability, while other materials rarely reflected this aspect. On the whole, the obtained Li[Ni_0.6_Co_0.2_Mn_0.2_]_0.98_Ga_0.02_O_2_ in this work exhibits excellent electrochemical and thermal safety, and is a very promising cathode material.

## 4. Conclusions

Ga-doped nickel-rich ternary layered Li[Ni_0.6_Co_0.2_Mn_0.2_]_1−x_Ga_x_O_2_ (x = 0, 0.01, 0.02, 0.03, 0.05) materials can be successfully prepared at 900 °C using the high-temperature solid-state reaction method. When the Ga content is 0.02, it is not only beneficial for promoting grain growth, but it can also reduce the charge transfer resistance and improve the interface structure and bulk structure stability, thereby significantly improving the electrochemical performance and thermal safety of NCM622 material. Compared with the pristine electrode, the Li(Ni_0.6_Co_0.2_Mn_0.2_)_0.98_Ga_0.02_O_2_ electrode exhibits remarkably improved electrochemical performance and thermal safety. At 0.5 C rate, the discharge capacity increases from 169.3 to 177 mAh g^−1^, and the capacity retention after 50 cycles also rises from 82.8% to 89.8%. In the charged state of 4.3 V, its exothermic temperature increases from 245.13 °C to more than 271.24 °C, and the total exothermic heat decreases from 561.7 to 225.6 J·g^−1^. Compared with some doped NCM622 materials reported in recent literature, the obtained Li[Ni_0.6_Co_0.2_Mn_0.2_]_0.98_Ga_0.02_O_2_ in this work exhibits excellent electrochemical and thermal safety and is a very promising cathode material. Enhancing the structural stability of NCM622 material in the charge–discharge process by Ga doping improves cycle stability and thermal safety, which can provide a new idea for improving the performance of long-life high-safety nickel-rich ternary materials for lithium-ion batteries.

## Figures and Tables

**Figure 1 materials-14-01816-f001:**
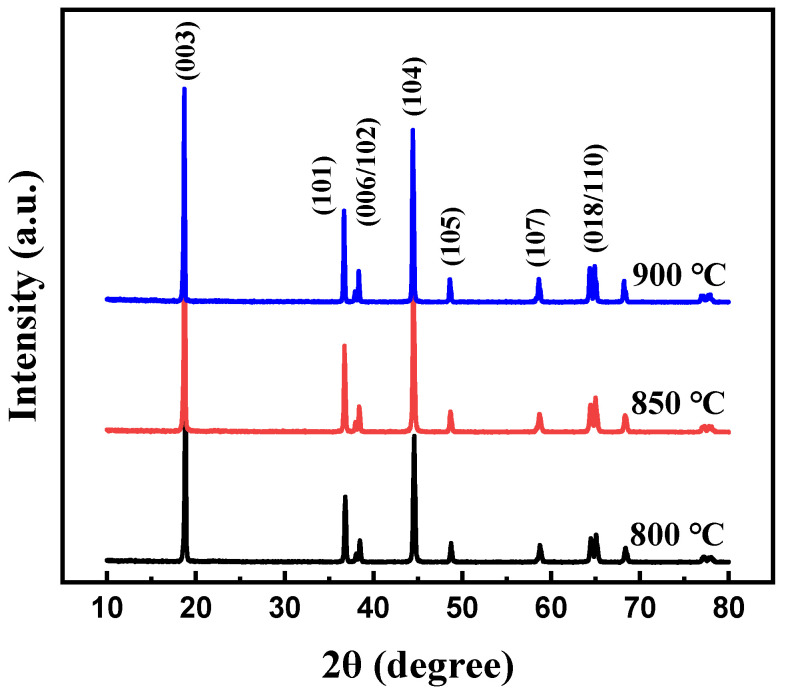
XRD patterns of Li[Ni_0.6_Co_0.2_Mn_0.2_]_0.98_Ga_0.02_O_2_ calcinated at different temperatures.

**Figure 2 materials-14-01816-f002:**
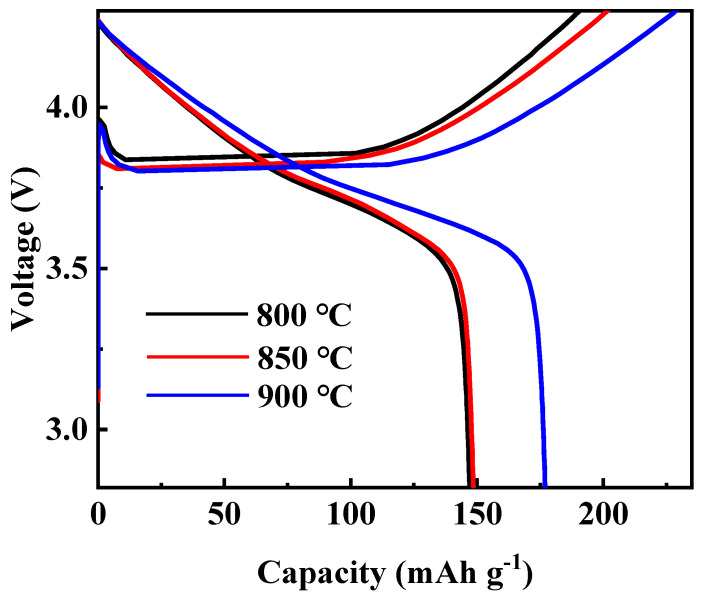
Initial charge–discharge curves of Li[Ni_0.6_Co_0.2_Mn_0.2_]_0.98_Ga_0.02_O_2_ calcinated at different temperatures.

**Figure 3 materials-14-01816-f003:**
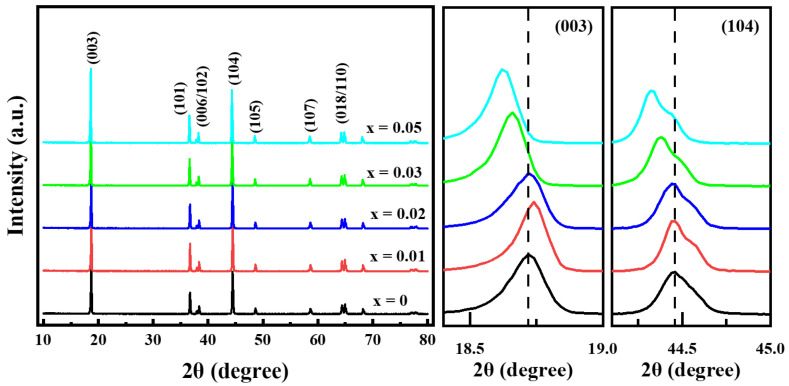
XRD patterns of Li[Ni_0.6_Co_0.2_Mn_0.2_]_1−x_Ga_x_O_2_ samples.

**Figure 4 materials-14-01816-f004:**
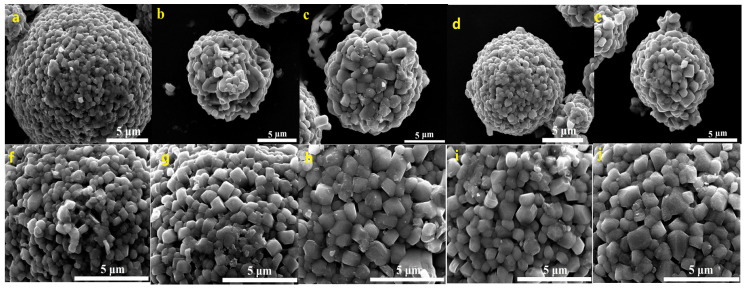
FESEM images of Li[Ni_0.6_Co_0.2_Mn_0.2_]_1−x_Ga_x_O_2_: x = 0 (**a**,**f**), x = 0.01 (**b**,**g**), x = 0.02 (**c**,**h**), x = 0.03 (**d**,**i**), x = 0.05 (**e**,**j**).

**Figure 5 materials-14-01816-f005:**
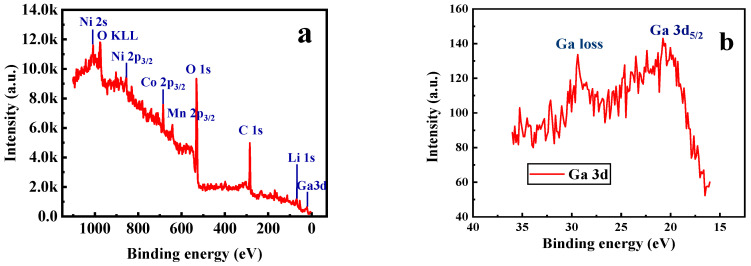
XPS spectra of Li[Ni_0.6_Co_0.2_Mn_0.2_]_0.98_Ga_0.02_O_2_. (**a**) Survey scan; (**b**) Ga 3d spectra.

**Figure 6 materials-14-01816-f006:**
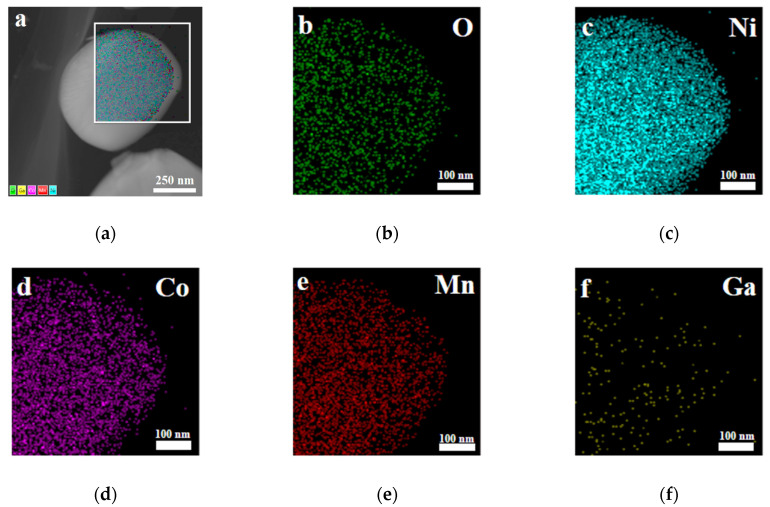
Energy dispersive X-ray (EDX) spectroscopy mapping of Li[Ni_0.6_Co_0.2_Mn_0.2_]_0.98_Ga_0.02_O_2_. (**a**) Layered image; (**b**) O atoms; (**c**) Ni atoms; (**d**) Co atoms; (**e**) Mn atoms; (**f**) Ga atoms.

**Figure 7 materials-14-01816-f007:**
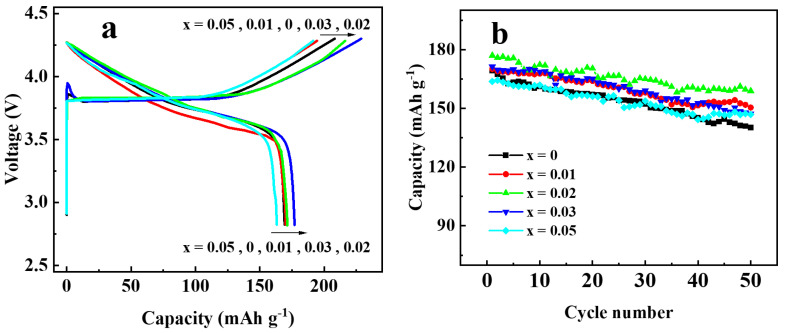
Charge–discharge performance of Li[Ni_0.6_Co_0.2_Mn_0.2_ ]_1−x_Ga_x_O_2_ at 0.5C rate: (**a**) initial charge–discharge curves; (**b**) cycling performance.

**Figure 8 materials-14-01816-f008:**
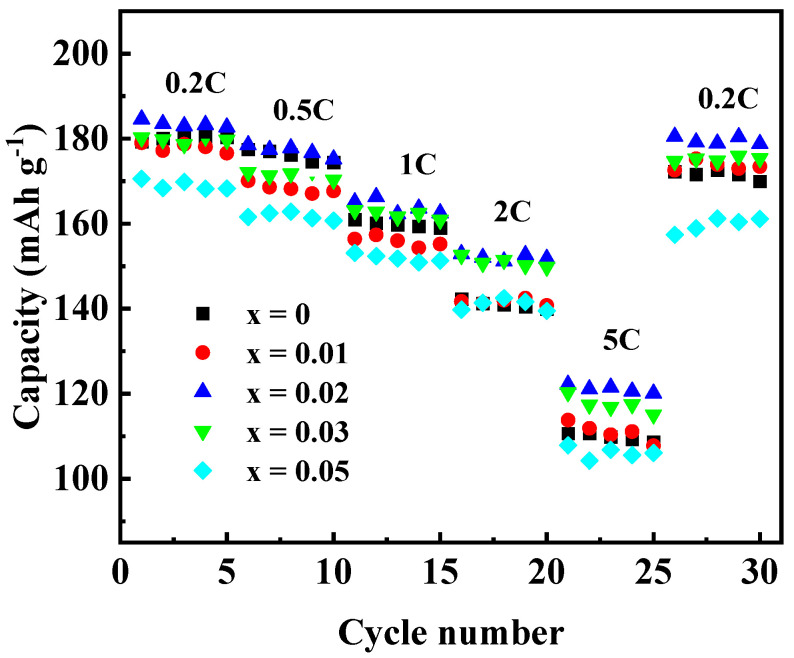
Rate capabilities of Li[Ni_0.6_Co_0.2_Mn_0.2_]_1−x_Ga_x_O_2_.

**Figure 9 materials-14-01816-f009:**
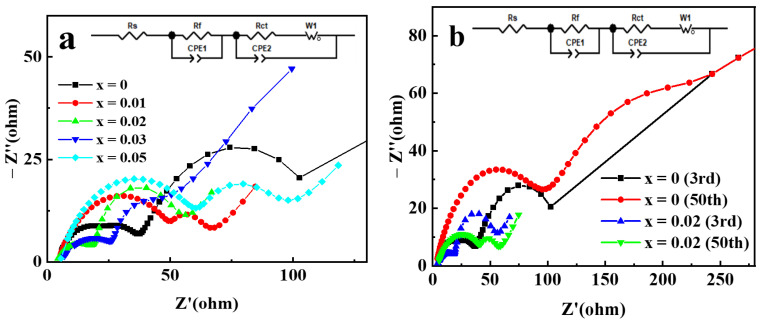
The Nyquist plots of Li[Ni_0.6_Co_0.2_Mn_0.2_]_1−x_Ga_x_O_2_ electrodes: (**a**) after 3 cycles; (**b**) comparison after 50 cycles with 3 cycles for x = 0 and 0.02 electrodes.

**Figure 10 materials-14-01816-f010:**
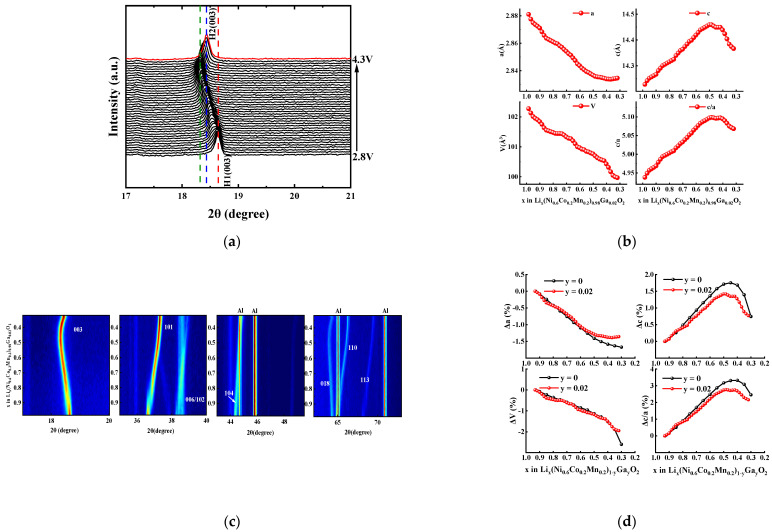
In situ XRD analysis for Li[Ni_0.6_Co_0.2_Mn_0.2_]_0.98_Ga_0.02_O_2_ electrode during charging process from 2.8 V to 4.3 V: (**a**) (003) diffraction peaks; (**b**) contour plots; (**c**) changes in lattice parameters; (**d**) relative change of lattice parameters (the data for sample with y = 0 from reference [5]).

**Figure 11 materials-14-01816-f011:**
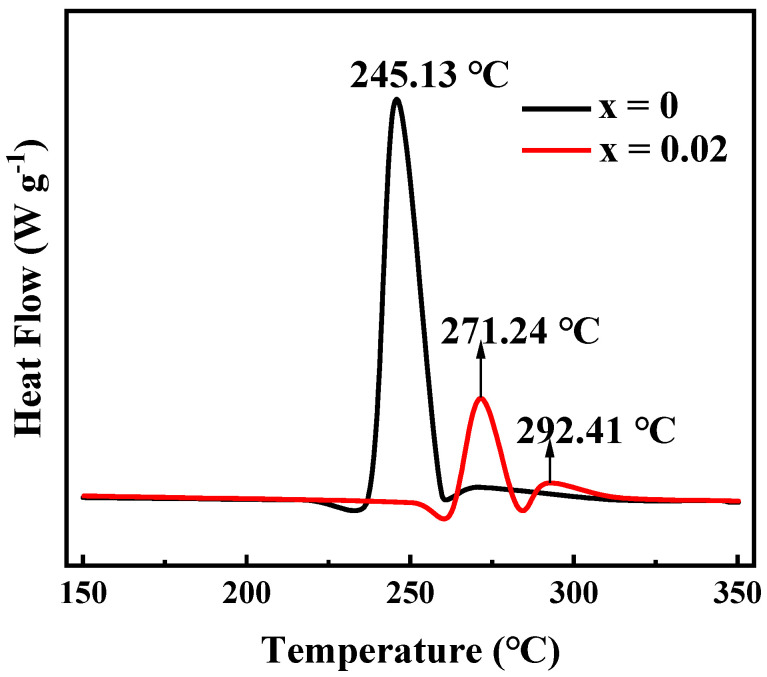
Differential scanning calorimeter (DSC) curves of Li[Ni_0.6_Co_0.2_Mn_0.2_]_1−x_Ga_x_O_2_ (x = 0, 0.02) charged to 4.3 V.

**Table 1 materials-14-01816-t001:** Electrochemical impedance spectroscopy (EIS) fitting results of Li[Ni_0.6_Co_0.2_Mn_0.2_]_1−x_Ga_x_O_2_ electrodes after 3 cycles.

x	R_s_/Ω	R_f_*/*Ω	R_ct_/Ω	(R_f_ + R_ct_)/Ω
0	4.531	32.30	82.47	114.8
0.01	4.658	48.82	17.92	66.74
0.02	3.532	18.62	32.34	50.96
0.03	5.682	23.58	36.19	59.77
0.05	5.229	59.11	38.87	97.98

**Table 2 materials-14-01816-t002:** Comparison of electrochemical performance of doped LiNi_0.6_Co_0.2_Mn_0.2_O_2_ materials from different research.

Doped Element	Preparation Method	Voltage Range (V)	Discharge Capacity (mAh g^−1^)	Capacity Retention	Thermal Stability	Ref.
Ga	Solid-state method	2.8–4.3	183.4 (0.2 C) 177 (0.5 C) 166 (1 C) 152.7 (2 C) 121.1 (5 C)	82.8% (undoped) and 89.8% (doped) (50 cycles, 0.5 C)	Remarkable improved	This work
Zr	Self-combustion synthesis	2.8–4.3	~160 (0.1 C) ~145 (C/3)	88.3% (undoped) and 93.1% (doped) (45 cycles, C/3)	No test	Ref. [22]
Na	Solid-state method	2.8–4.3	176 (0.2 C) 170 (0.5 C) 162 (1 C)	83.7% (undoped) and 93.5% (doped) (100 cycles, 1 C)	No test	Ref. [24]
Mg	Solid-state method	2.8–4.3	177.07 (0.1 C) 162.6 (1 C)	79.33% (undoped) and 90.02% (doped) (100 cycles, 1 C)	No test	Ref. [25]
F	Solid-state method	2.5–4.3	163.5 (0.1 C) 146.1 (1 C)	89.2% (undoped) and 94.2% (doped) (50 cycles, 1 C)	No test	Ref. [27]
Na + F	Solid-state method	2.7–4.3	171 (0.1 C) 141 (1 C)	87% (doped) (100 cycles, 1 C)	No test	Ref. [28]

## Data Availability

The data presented in this study are available on request from the corresponding author.

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
