# Peer review of "Enhanced Structural Stability and Electrochemical Performance of LiNi_0.6_Co_0.2_Mn_0.2_O_2_ Cathode Materials by Ga Doping"

_materials, 2021, doi:10.3390/ma14081816_

Round 1

Reviewer 1 Report

The work is well structured. The topic addressed is a topical one. The introduction provides a good, generalized background of the topic that quickly gives the reader an appreciation of the wide range of applications for performance of nickel-rich ternary cathode materials for Li-ion batteries.

The methods used are appropriate to the aims of the study.

The information provided is sufficient for a capable researcher to reproduce the experiments described.

I do not think any additional experiments are required to validate the results of those that were performed. though I don’t think this is vital to the present paper (especially given the length limitations on the paper), but it may be something that would be helpful in a longer, more detailed paper: Cyclic voltammetry on the studied electrodes.

The conclusions summarize well the experimental results

I want to congratulate authors for their research.

Author Response

Reviewer 1

Comments and Suggestions for Authors:

The work is well structured. The topic addressed is a topical one. The introduction provides a good, generalized background of the topic that quickly gives the reader an appreciation of the wide range of applications for performance of nickel-rich ternary cathode materials for Li-ion batteries.

The methods used are appropriate to the aims of the study.

The information provided is sufficient for a capable researcher to reproduce the experiments described.

I do not think any additional experiments are required to validate the results of those that were performed. though I don’t think this is vital to the present paper (especially given the length limitations on the paper), but it may be something that would be helpful in a longer, more detailed paper: Cyclic voltammetry on the studied electrodes.

The conclusions summarize well the experimental results

I want to congratulate authors for their research.

Reviewer 2 Report

This manuscript, “Enhanced Structural Stability and Electrochemical Performance of LiNi0.6Co0.2Mn0.2O2 Cathode Materials by Ga-doping”, reports the results of Ga-doping on a widely used lithium-ion battery cathode material, NMC622. The authors have conducted sufficient material characterization to confirm Ga doping and the required performance analysis to compare with the undoped cathode. The results appear interesting and noteworthy. However, there are a few minor concerns as follows, that need to be addressed before it could be published in Materials:

  1. While it is always interesting to investigate the effects of using new metals for doping, the question about their economics naturally arises. And since NMC622 is already being used by commercial players, it is all the more important to weigh the cost and benefits. Could the authors include a discussion on the geographical abundance and the cost of gallium precursor they have used for the doping, in the Introduction section? This would help the readers and market leaders get some commercial context.
  2. For the x=0.02 sample in Figure 5a, there seems to be a small overpotential (of ~ 0.1V) at the beginning of the charge step before the onset of the plateau (i.e., between 3.8 V and 4.0 V). Could the authors comment on the possible reason(s) for this?
  3. Does this overpotential occur only for x=0.02? It is not exactly clear since the charge profiles seem to overlap for all values of x.
  4. In Figure 7b, for the x=0.02 sample, the charge transfer resistance (mid-frequency semicircle) decreases after 50 cycles. This is in contrast to the trend observed in the undoped (x=0) sample, and therefore, it further cements the rationale behind Ga-doping. Could the authors elaborate (or) hypothesize on why the charge transfer resistance decreases after 50 cycles?
  5. As I understand, the highlight of this work is the performance enhancement achieved with such low amount of Ga-doping (x=0.02). Although the authors have covered sufficient previous literature on doping other elements (F, Mg, Zr etc.) in the Introduction section, it would be good if the authors could specify more details such as the extent of doping (x=?) and the performance improvement (in initial capacity, capacity retention % etc.) achieved in those prior works. Specifying these quantities and providing a quantitative comparison would help emphasize the merits of this work.

Author Response 

Reviewer 2

Comments and Suggestions for Authors:

This manuscript, “Enhanced Structural Stability and Electrochemical Performance of LiNi0.6Co0.2Mn0.2O2 Cathode Materials by Ga-doping”, reports the results of Ga-doping on a widely used lithium-ion battery cathode material, NMC622. The authors have conducted sufficient material characterization to confirm Ga doping and the required performance analysis to compare with the undoped cathode. The results appear interesting and noteworthy. However, there are a few minor concerns as follows, that need to be addressed before it could be published in Materials:

  1. While it is always interesting to investigate the effects of using new metals for doping, the question about their economics naturally arises. And since NMC622 is already being used by commercial players, it is all the more important to weigh the cost and benefits. Could the authors include a discussion on the geographical abundance and the cost of gallium precursor they have used for the doping, in the Introduction section? This would help the readers and market leaders get some commercial context.

Response to comment: We thank the reviewer for the careful review and constructive suggestions on our manuscript. We appreciate the opportunity to address the reviewer’s concern. According to the reviewer’s suggestion, a discussion on the geographical abundance and the cost of gallium has been added in the introduction section. The reference [38] was also added in the revised manuscript. The revised text is as follows:

Although Ga is a dispersed rare metal, its reserves on the earth have reached 1 million tons, which is about 1/7 of that of cobalt, but the price is less than 4 times that of cobalt [38]. Moreover, the amount of doped Ga is very small, the cost increase should be minimal. Therefore, the research on Ga doping to improve the electrochemical performance of nickel-rich LiNi1-x-yCoxMnyO2 materials is still a necessary work.

  1. For the x=0.02 sample in Figure 5a, there seems to be a small overpotential (of ~ 0.1V) at the beginning of the charge step before the onset of the plateau (i.e., between 3.8 V and 4.0 V). Could the authors comment on the possible reason(s) for this?

Response to comment: We thank the reviewer for pointing this out. We appreciate the opportunity to address the reviewer’s concern. In the first cycle, a small overpotential (of ~ 0.1V) at the beginning of the charge step before the onset of the plateau was often observed in some reported literatures, e.g. Fig.6f in the reference (Nanoscale, 2018,10, 8820-8831).

Especially, this phenomenon is more common when the charging current is large. It is because that ternary cathode materials LiNi1-x-yCoxMnyO2 are semiconductors. At the beginning of the charge step before the onset of the plateau, it has high impedance, small Li+ diffusion coefficient, and thus high overpotential, e.g. Fig.9 in the reference (Adv. Energy Mater. 2017, 1701788).

In this work, the charge current in the first cycle was 0.5C, which was large, and thus small overpotential appeared.

  1. Does this overpotential occur only for x=0.02? It is not exactly clear since the charge profiles seem to overlap for all values of x.

Response to comment: We thank the reviewer for pointing this out. We appreciate the opportunity to address the reviewer’s concern. It can be seen that this overpotential occur for all the four samples. The reason is the same as above.

  1. In Figure 7b, for the x=0.02 sample, the charge transfer resistance (mid-frequency semicircle) decreases after 50 cycles. This is in contrast to the trend observed in the undoped (x=0) sample, and therefore, it further cements the rationale behind Ga-doping. Could the authors elaborate (or) hypothesize on why the charge transfer resistance decreases after 50 cycles?

Response to comment: We thank the reviewer for pointing this out. We appreciate the opportunity to address the reviewer’s concern. In the reported literatures, there are also the phenomenon that the charge transfer impedance decreases after cycling, and even the SEI film impedance also decreases. For example, in the reference (Journal of The Electrochemical Society, 2017, 164(2), A475-A481.), for Al2O3-coated NCM sample in the Table I, the Rsf and Rct values decreased from the 10 cycle to the 30 cycle.

The same phenomenon also appeared for LiNi0.8Co0.1Mn0.1O2 cathode material in the reference (Electrochimica Acta, 2019, 309, 74-85). As shown in Table 2 and Table 3, 3, the RSEI and Rct values also decreased after 100 cycles at 1C and 5C rate. However, there are few literatures discussing the reasons for the decrease.

Actually, in the initial stage of the charge-discharge cycling, the impedance of the test cell often gradually decreases due to the electrode activation. After some cycles, the impedance gradually rises again due to the electrode/electrolyte interface side reaction (e.g. electrolyte decomposition, formation of SEI film and transition metal dissolution) and the structural instability (e.g. cation mixing, NiO phase formation, microcracks and microstrains caused by phase transition, etc.). In this work, the impedance between the 3rd cycle and the 50th cycle are compared. We speculate that the electrode activation may not be complete after 3 cycles, that is, the charge transfer impedance is not at the lowest value. For the pristine electrode (x=0), large increase in impedance after 50 cycles ensure an increasing trend compared to the 3rd cycle. However, for the doped material (x=0.02), due to its better structural stability, the impedance increases slowly after electrode activation, and the increased impedance value after 50 cycles is still not as large as the decreased impedance value from the 3rd cycle to the end of electrode activation, thus showing a decreased trend compared to the 3rd cycle.

  1. As I understand, the highlight of this work is the performance enhancement achieved with such low amount of Ga-doping (x=0.02). Although the authors have covered sufficient previous literature on doping other elements (F, Mg, Zr etc.) in the Introduction section, it would be good if the authors could specify more details such as the extent of doping (x=?) and the performance improvement (in initial capacity, capacity retention % etc.) achieved in those prior works. Specifying these quantities and providing a quantitative comparison would help emphasize the merits of this work.

Response to comment: We appreciate the reviewer’s constructive suggestions and comments. According to the reviewer’s suggestions, the performance of some doped LiNi0.6Co0.2Mn0.2O2 materials reported in recent years are listed in Table 2, and the relative discussion has also been added in the revised manuscript. The revised text is as follows:

Table 2. lists the performance of some doped LiNi0.6Co0.2Mn0.2O2 materials reported in recent years. These materials are selected because they have the same charge cut-off voltage, which is convenient for performance comparison. It can be seen that the Ga-doped sample synthesized in this work exhibits the highest discharge capacity and rate performance. Comparing the cycle performance, although the capacity retention of some samples seems to be a little higher, but it is obtained under the circumstance of high rate (1C) and low initial capacity, which is not feasible for comparison. It is inferred that the cycle performance of Ga-doped sample should be comparable. In addition, the Ga-doped sample shows significantly improved thermal stability, while other materials rarely consider this aspect. On the whole, the obtained Li[Ni0.6Co0.2Mn0.2]0.98Ga0.02O2 in this work exhibits excellent electrochemical and thermal safety, and is a very promising cathode material.

Table 2. Comparison of electrochemical performance of doped LiNi0.6Co0.2Mn0.2O2 materials from different research

Doped element

Preparation method

Voltage range (V)

Discharge

capacity

 (mAh g-1)

Capacity retention

Thermal

stability

Ref.

Ga

Solid state method

2.8-4.3

183.4 (0.2C)

177 (0.5C)

166 (1C)

152.7 (2C)

121.1 (5C)

82.8% (undoped) and

89.8% (doped)

(50 cycles, 0.5C)

Remarkable improved

This work

Zr

Self-combustion synthesis

2.8-4.3

~160 (0.1C)

~145 (C/3)

88.3% (undoped) and

 93.1% (doped)

 (45 cycles, C/3 )

No test

[22]

Na

Solid state method

2.8-4.3

176 (0.2C)

170 (0.5C)

162 (1C)

83.7% (undoped) and

93.5% (doped)

(100 cycles, 1C)

No test

[24]

Mg

Solid state method

2.8-4.3

177.07 (0.1C)

162.6 (1C)

79.33% (undoped) and

90.02% (doped)

(100 cycles, 1C)

No test

[25]

F

Solid state method

2.5-4.3

163.5 (0.1C)

146.1 (1C)

89.2% (undoped) and

 94.2% (doped)

(50 cycles, 1C)

No test

[27]

Na + F

Solid state method

2.7-4.3

171 (0.1C)

141 (1C)

87% (doped)

(100 cycles, 1C)

No test

[28]

In addition, a sentence has been added in the conclusions to demonstrate clearly the significance of this work. The revised text is as follows:

Compared with some doped NCM622 materials reported in recent literatures, the obtained Li[Ni0.6Co0.2Mn0.2]0.98Ga0.02O2 in this work exhibits excellent electrochemical and thermal safety, and is a very promising cathode material. Enhancing the structural stability of NCM622 material in the charge-discharge process by Ga doping, thereby improving cycle stability and thermal safety, which can provide a new idea for improving the performance of long-life, high-safety nickel-rich ternary materials for lithium-ion batteries.

Reviewer 3 Report

In this manuscript, the authors proposed a promising route to improve the electrochemical performances of NCM cathode by Ga-doping. The originality of the data is appropriate. The results and findings are potentially interesting for electrochemists and readers in such community. The authors are suggested to address some questions before possible acceptance for publication, as specified below.

  1. First of all, I suggest to authors to compare the structural parameters of material before and after Ga doping in order to clarify its effect on expanding or contraction the unit cell.
  2. I suggest to evaluate the chemical diffusion coefficients from EIS data to clarify the relationship between doping and opening of structure.
  3. Some experiments are required to confirm the existence of Ga in material, may be XPS or so on. Besides, EDX mapping is suggested to perform in order to confirm the uniform doping of material.
  4. Significance of these Ga-doped NCM cathodes should be clearly demonstrated. The authors should summarize the results (may be in tabular form) and compare the main tested parameters with that from recent papers on modified NCM.

As I believe, the revised manuscript that addresses these concerns may be acceptable for publication in the Materials.

Author Response

Reviewer 3

Comments and Suggestions for Authors

In this manuscript, the authors proposed a promising route to improve the electrochemical performances of NCM cathode by Ga-doping. The originality of the data is appropriate. The results and findings are potentially interesting for electrochemists and readers in such community. The authors are suggested to address some questions before possible acceptance for publication, as specified below.

  1. First of all, I suggest to authors to compare the structural parameters of material before and after Ga doping in order to clarify its effect on expanding or contraction the unit cell.

Response to comment: We thank the reviewer for the careful review and constructive suggestions on our manuscript. We appreciate the opportunity to address the reviewer’s concern. Actually, the lattice parameters of material before [5] and after Ga doping (x=0.02) has been compared in the original manuscript. In order to express this more clearly, the revised manuscript further uses graphics (Fig. 10d) to show the influence of Ga doping on the relative change of lattice parameters during the delithiation process. It is clear that the relative change of lattice parameters can be suppressed by Ga doping during the charging process, which is obviously beneficial to the cycle performance of the material. The relative discussion has also been revised, and the revised text is as follows:

As shown in Fig. 10d, the relative change of lattice parameters of Li[Ni0.6Co0.2Mn0.2]0.98Ga0.02O2 are compared with the reported values of undoped NCM622 material [5] during the delithiation process. It can be seen that----

  1. I suggest to evaluate the chemical diffusion coefficients from EIS data to clarify the relationship between doping and opening of structure.

Response to comment: We thank the reviewer for pointing this out. We appreciate the opportunity to address the reviewer’s concern. According to the reviewer’s suggestions, the Li diffusion coefficients were calculated from EIS data by the method in the reference [28], and the relative discussion has also been added in the revised manuscript. The revised text is as follows:

The Li+ diffusion coefficient can also be calculated by using the method in the reference [28] to process the EIS data. The obtained Li+ diffusion coefficients for the samples with Ga content x=0, 0.01, 0.02, 0.03, and 0.05 are 1.30×10-12,5.80×10-11,8.85×10-11,6.92×10-11,and 3.81×10-11 cm2×s-1, respectively. The change trend of the Li+ diffusion coefficient is completely consistent with that of the charge transfer impedance, which should be related to the synergy of charge transfer. Compared with the pristine sample, the Ga-doped sample with x=0.02 exhibits a Li+ diffusion coefficient as high as about 53 times, which is consistent with its best electrochemical performance.

  1. Some experiments are required to confirm the existence of Ga in material, may be XPS or so on. Besides, EDX mapping is suggested to perform in order to confirm the uniform doping of material.

Response to comment: We appreciate the reviewer’s constructive suggestions and comments. According to the reviewer’s suggestions, XPS and EDX mapping analysis have been provided in the revised manuscript. XPS confirmed the presence of Ga3+ in the sample, and EDX mapping confirmed that Ga3+ ions were uniformly doped into the material. The relative experimental and discussion have also been revised completely, and the relative reference [38] was also added. The revised text is as follows:

Furthermore, X-ray photoelectron spectroscopy (XPS) and Energy Dispersive X-Ray Spectroscopy (EDX) analysis were carried out for the sample with x=0.02. As shown in Fig. 5a, the photoelectron peak of Ga 3d can be observed in the XPS survey spectra. It can be seen from the Fig. 5b that the binding energy corresponding to the Ga 3d5/2 peak is 20.8 eV, which is consistent with the reported data for Ga2O3 [38], indicating that the valence state of doped Ga ions remains +3. Fig. 6 presents the EDX mapping of Li[Ni0.6Co0.2Mn0.2]0.98Ga0.02O2. Same as Ni, Co, Mn, and O atoms, the doped Ga element is also uniformly distributed, which can conform uniform Ga doping in the material.

  1. Significance of these Ga-doped NCM cathodes should be clearly demonstrated. The authors should summarize the results (may be in tabular form) and compare the main tested parameters with that from recent papers on modified NCM.

Response to comment: We appreciate the reviewer’s constructive suggestions and comments. According to the reviewer’s suggestions, the performance of some doped LiNi0.6Co0.2Mn0.2O2 materials reported in recent years are listed in Table 2, and the relative discussion has also been added in the revised manuscript. The revised text is as follows:

Table 2. lists the performance of some doped LiNi0.6Co0.2Mn0.2O2 materials reported in recent years. These materials are selected because they have the same charge cut-off voltage, which is convenient for performance comparison. It can be seen that the Ga-doped sample synthesized in this work exhibits the highest discharge capacity and rate performance. Comparing the cycle performance, although the capacity retention of some samples seems to be a little higher, but it is obtained under the circumstance of high rate (1C) and low initial capacity, which is not feasible for comparison. It is inferred that the cycle performance of Ga-doped sample should be comparable. In addition, the Ga-doped sample shows significantly improved thermal stability, while other materials rarely consider this aspect. On the whole, the obtained Li[Ni0.6Co0.2Mn0.2]0.98Ga0.02O2 in this work exhibits excellent electrochemical and thermal safety, and is a very promising cathode material.

Table 2. Comparison of electrochemical performance of doped LiNi0.6Co0.2Mn0.2O2 materials from different research

Doped element

Preparation method

Voltage range (V)

Discharge

capacity

 (mAh g-1)

Capacity retention

Thermal

stability

Ref.

Ga

Solid state method

2.8-4.3

183.4 (0.2C)

177 (0.5C)

166 (1C)

152.7 (2C)

121.1 (5C)

82.8% (undoped) and

89.8% (doped)

(50 cycles, 0.5C)

Remarkable improved

This work

Zr

Self-combustion synthesis

2.8-4.3

~160 (0.1C)

~145 (C/3)

88.3% (undoped) and

 93.1% (doped)

 (45 cycles, C/3 )

No test

[22]

Na

Solid state method

2.8-4.3

176 (0.2C)

170 (0.5C)

162 (1C)

83.7% (undoped) and

93.5% (doped)

(100 cycles, 1C)

No test

[24]

Mg

Solid state method

2.8-4.3

177.07 (0.1C)

162.6 (1C)

79.33% (undoped) and

90.02% (doped)

(100 cycles, 1C)

No test

[25]

F

Solid state method

2.5-4.3

163.5 (0.1C)

146.1 (1C)

89.2% (undoped) and

 94.2% (doped)

(50 cycles, 1C)

No test

[27]

Na + F

Solid state method

2.7-4.3

171 (0.1C)

141 (1C)

87% (doped)

(100 cycles, 1C)

No test

[28]

 In addition, a sentence has been added in the conclusions to demonstrate clearly the significance of this work. The revised text is as follows:

Compared with some doped NCM622 materials reported in recent literatures, the obtained Li[Ni0.6Co0.2Mn0.2]0.98Ga0.02O2 in this work exhibits excellent electrochemical and thermal safety, and is a very promising cathode material. Enhancing the structural stability of NCM622 material in the charge-discharge process by Ga doping, thereby improving cycle stability and thermal safety, which can provide a new idea for improving the performance of long-life, high-safety nickel-rich ternary materials for lithium-ion batteries.

Reviewer 4 Report

The upshot of this manuscript is that the synthesis and characterization of Ga-doped NCM622 materials. The effect of annealing temperature, Ga content on the structural and electrochemical properties of Ga-doped NCM622 materials are studied. The Ga-doped material prepared under the optimized synthesis conditions present improved structural stability and electrochemical performance. I would recommend this paper to be published in Materials if the authors can address the following comments:

  • Did the authors try to calcine at 950 or 1000 oC to confirm 900 oC was the optimum temperate?
  • Could the authors please explain the nature of the second peak at 292.41 oC?
  • Did the authors use the same amount of the material for DSC measurements in order to compare the results at x = 0 and x = 0.02?

Author Response

Reviewer 4

Comments and Suggestions for Authors

The upshot of this manuscript is that the synthesis and characterization of Ga-doped NCM622 materials. The effect of annealing temperature, Ga content on the structural and electrochemical properties of Ga-doped NCM622 materials are studied. The Ga-doped material prepared under the optimized synthesis conditions present improved structural stability and electrochemical performance. I would recommend this paper to be published in Materials if the authors can address the following comments:

  1. Did the authors try to calcine at 950 or 1000 ℃ to confirm 900 ℃ was the optimum temperate?

Response to comment: We thank the reviewer for the careful review and constructive suggestions on our manuscript. We appreciate the opportunity to address the reviewer’s concern. In the literatures [22-31], the calcination temperature has been reported to be 800-900 ℃ for synthesis of ion-doped NCM622 materials, and there are few reports about calcination at 950 ℃. Considering the doped Ga amount is very small, the optimized calcination temperature range was also selected at 800-900 ℃ in this work. The experimental results also confirmed that the sample calcined at 900 ℃ exhibits a well-ordered layered structure and high discharge capacity. Therefore, in the work, samples calcined at 900 °C are used to discuss the influence of Ga doping on the structure and electrochemical performance of Li[Ni0.6Co0.2Mn0.2]1-xGaxO2 materials.

  1. Could the authors please explain the nature of the second peak at 292.41 oC?

Response to comment: We appreciate the reviewer’s constructive suggestions and comments. As we know, similar phenomena for the delithiated layered cathode materials has also been reported in literatures, for example, LiCoO2 (LCO) and LiCoO2@1% LiNi0.45Al0.05Mn0.5O2 (LCO@1%LNAMO) (Fig.5b, Electrochimica Acta, 2019, 327, 135018), and LiNi1-x-yCoxMnyO2 (Fig.12, Journal of The Electrochemical Society, 2007, 154(10), A971-A977).

In addition to the main exothermic peak attributed to phase transition, there may be one or two small peaks. This is related to the side reaction between the cathode material and electrolyte (Electrochimica Acta, 2019, 327, 135018). In the revised manuscript, the nature of the second peak at 292.41 was explained, and the relative reference [40] was also added. The revised text is as follows:

“However, for the Ga-doped sample Li[Ni0.6Co0.2Mn0.2]0.98Ga0.02O2, in addition to an exothermic peak attributed to the phase transition at 271.24 ℃, a small exothermic peak also appeared at 292.41 ℃, which may be related to the side reaction between cathode and electrolyte [39]. It is clear that not only does the exothermic peak temperature increase, but the total exothermic heat also decreases to 225.6 J·g-1.”

  1. Did the authors use the same amount of the material for DSC measurements in order to compare the results at x = 0 and x = 0.02?

Response to comment: We thank the reviewer for pointing this out. We appreciate the opportunity to address the reviewer’s concern. In this work, the same amount (typical weight 6.2 mg) of the material for DSC measurements was used in order to accurately compare the results at x = 0 and x = 0.02. In order to express this clearly, the typical weight was added in the experimental of the revised manuscript. It is “The thermal stability of Li[Ni0.6Co0.2Mn0.2]1-xGaxO2 materials (typical weight 6.2 mg) in the charged state (4.3 V, vs. Li/Li+) was analyzed--------”.

Round 2

Reviewer 3 Report

I think that this manuscript had been improved significantly and now it can be accepted.